# Cellular and Molecular Triggers of Retinal Regeneration in Amphibians

**DOI:** 10.3390/life13101981

**Published:** 2023-09-28

**Authors:** Yuliya V. Markitantova, Eleonora N. Grigoryan

**Affiliations:** Koltzov Institute of Developmental Biology, Russian Academy of Sciences, 119334 Moscow, Russia; yuliya.mark@gmail.com

**Keywords:** amphibia, eye, retina, regeneration, cell sources, injury-induced, regenerative responses

## Abstract

Understanding the mechanisms triggering the initiation of retinal regeneration in amphibians may advance the quest for prevention and treatment options for degenerating human retina diseases. Natural retinal regeneration in amphibians requires two cell sources, namely retinal pigment epithelium (RPE) and ciliary marginal zone. The disruption of RPE interaction with photoreceptors through surgery or injury triggers local and systemic responses for retinal protection. In mammals, disease-induced damage to the retina results in the shutdown of the function, cellular or oxidative stress, pronounced immune response, cell death and retinal degeneration. In contrast to retinal pathology in mammals, regenerative responses in amphibians have taxon-specific features ensuring efficient regeneration. These include rapid hemostasis, the recruitment of cells and factors of endogenous defense systems, activities of the immature immune system, high cell viability, and the efficiency of the extracellular matrix, cytoskeleton, and cell surface remodeling. These reactions are controlled by specific signaling pathways, transcription factors, and the epigenome, which are insufficiently studied. This review provides a summary of the mechanisms initiating retinal regeneration in amphibians and reveals its features collectively directed at recruiting universal responses to trauma to activate the cell sources of retinal regeneration. This study of the integrated molecular network of these processes is a prospect for future research in demand biomedicine.

## 1. Introduction

Amphibians are a class of animals on the evolutionary ladder that exhibit a wide range of regenerative abilities. Members of this class can regrow lost or damaged body parts: organs, tissues, and even complex structures such as limbs, tail, spinal cord, and parts of the heart and brain [1,2,3,4,5,6]. The new retina of the eye formed as a result of regeneration in mature amphibians is a rare example that could be a desirable scenario in the case of retinal degeneration following after retinal damage and disease. 

In humans, the major retinal disorders are age-related macular degeneration (AMD), glaucoma, retinitis pigmentosa (RP), retinal detachment, and proliferative vitreoretinopathy (PVR) [7,8,9,10,11,12,13]. These disorders are caused by the loss of cells and cell-to-cell interactions in the functional light perception system, the retinal pigmented epithelium (RPE) ↔ neural retina (NR). In humans, due to the lack of ability to regenerate damaged retina, these disorders lead to impaired vision and, in extreme cases, blindness. The available approaches that have been used to date in the attempt to restore human vision include drug and gene therapy, cell and tissue transplantation, and prosthetic stimulators. These procedures are, in fact, invasive and, even with the best outcomes, can only slow down the process of vision deterioration. 

Therefore, the examples of complete retinal regeneration by RPE and the ciliary marginal zone (CMZ) of the retina with the restoration of cells’ composition and functions in amphibians become even more valuable and essential in this study. All the species of Urodela amphibians studied at different times by different research teams (caudate amphibians: *Triturus viridescens*, *Triturus cristatus*, *Pleurodeles waltl*, *Notophtalmus viridescens*, and *Ambystoma mexicanum*, and Anura: *Xenopus laevis*, *Rana catasbiana*, *Rana tropicalis*) demonstrated the ability to regenerate the retina of the eye. The major features of regeneration observed within each of these two orders were the NR regeneration rate and the different contributions made by the cell sources, RPE and CMZ. Insights from amphibian species may hold important clues for induction and controlling restoration process in humans. Amphibians’ models of retinal regeneration have been studied for a long time, but the cellular and molecular mechanisms of the process remain largely elusive. The initiation of regeneration, the first event occurring shortly after retinal damage, largely determines the probability, means, and success of regeneration. The events initiating retinal regeneration in amphibians can be considered universal, generally characteristic of tissue regeneration in vertebrates. However, simultaneously, they have features only inherent only retinal regeneration-competent amphibians. These features are of major interest and are noted in the present review along with the discussion of common processes that usually accompany tissue regeneration. In the review, we briefly analyze cell death, the disruption of cell-to-cell communication, changes in cellular structure and extracellular matrix (ECM), oxidative stress (OS), responses from the circulatory and immune systems, as well as the known facts about the components of signaling regulatory networks. However, first, the modes of retinal regeneration in caudate and acaudate amphibians and the regeneration-inducing methods of retinal damage in amphibians should be briefly considered. 

## 2. Source Cells and Means of Retinal Regeneration

Caudate (Urodela) and acaudate (Anura) amphibians exhibit high regenerative potential, in particular for the regeneration of eye tissues. These animals have become popular models and vivid examples of the regeneration of the lens [14,15,16,17] and the retina [18,19,20,21,22,23,24] of the eye. In Urodela, which are the most regeneration-competent animals, the retina regenerates after various types of surgery (see below), including the surgical removal of the original neural retina (NR). In all types of operations, the RPE and the remaining cells of the extreme periphery of the retina, CMZ, become the source of a new, fully functioning retina [18,19,25,26,27,28,29,30,31] (Figure 1A). The regeneration of the retina observed after the NR removal in the mature newts of all the studied species begins with the displacement of RPE cells from the layer and the quick loss of the traits of the epithelial pigmented phenotype. These dedifferentiating cells proliferate and form a transitory population of cells resembling neuroblasts in morphology, high proliferative activity, and gene expression signature. The differentiation is initiated at 2–3 weeks post-operation when a progenitor cell population of the NR regenerate is formed and reaches the necessary number of cells. Then, the retinal anlage cells exit the reproduction cycle and differentiate into various retinal cell types. The NR regenerate is then stratified, and its function, as well as coordinated interaction with RPE, becomes established. The RPE cells, retained in the layer, bring their number in the RPE layer to a necessary level consistent with the emerging de novo NR and restore functional interactions with it through the proliferation and retention of the original features. CMZ cells participate in the retinal regeneration at the eye periphery. When regenerating the retina, caudate amphibians use many of the conserved mechanisms that were identified during the development of the retina in vertebrate embryogenesis in vivo. NR regeneration in Urodela occurs through the control of known developmental regulators: transcription factors (TFs) and signaling pathways. The RPE cells of non-operated animals have a melanocyte phenotype and express genes (*RPE65*, *Otx2*, *Mitf*, and *CRBP*) corresponding to this type of differentiation. The results of single-cell qPCR carried out on intact RPE cells did not show an expression of genes characteristic of stem cells or pro-neural progenitors [32]. A study of TFs expressed during the period of RPE reprogramming revealed a number of homeobox genes (*Pax6*, *Prox1*, *Six3*, *Pitx1*, and *Pitx2*) whose differential regulation was recorded during eye development in vertebrates. It is worth noting that, during the RPE dedifferentiation and cell type conversion, the expression of these TFs occurs against the background of expression of the tissue-specific *RPE65* and *Otx2* genes [27,33,34,35,36,37]. In particular, the up-regulation of the “developmental” TFs *Pax6*, *Six3*, as well as the genes encoding the signal molecule FGF2, was observed along with the *Otx2*, *RPE65*, and *CRBP* expression remaining at a relatively high level [27,35,36]. Several transcript variants of the *Pax6* gene, a key gene in the eye development, were found to encode protein isoforms [38]. The regulatory role of the differential expression of Pax6 isoforms is assumed in the process of the neural conversion of RPE. In addition to the expression of these TF genes and proteins, the reprogramming RPE cells is characterized by the expression of the *c-myc*, *Klf4*, and *Sox2* genes, which are markers of pluripotent cells. The expression of these genes confirms the low level of RPE-derived cell differentiation, and suggests the involvement of the *c-myc*, *Klf4*, and *Sox2* genes in the process of RPE cell reprogramming [32]. CMZ cells of the mature Urodela also comprise a population of low-differentiated cells having properties of progenitors. These cells are involved in NR regeneration, thus contributing to the NR formation and de novo growth on the periphery of the eye [39]. In newts *P. waltl*, *Pax6*, *Otx2*, *Six3*, *Prox1*, and *Pitx2* are the genes actively expressed in the CMZ cells of non-operated eyes and during retinal regeneration. This indicates that CMZ cells maintain and retain a low level of differentiation under normal conditions and throughout the NR regeneration process [33,34,36,37].

In addition to the above-indicated RPE and CMZ as the major sources of NR regeneration in newts, an alternative source of photoreceptor restoration was found: displaced bipolar-like, “underdifferentiated” cells in the inner nuclear layer of NR [40,41]. 

Retinal repair in vivo was also reported for several species of tailless amphibians, namely frogs [16,20,21,42,43,44]. In the model of Anura (*Xenopus laevis*), regeneration is possible after the removal of the original NR only if the retinal vascular membrane (RVM) is retained. The latter constitutes the inner limiting membrane of the NR and the reach of laminin [20]. In this case, RPE cells become displaced, leaving a layer, migrating towards RVM, dedifferentiating, settling on the RVM and proliferating, forming a population of neuroblasts (Figure 1B). This population of cells is the retina regenerate anlage which then develops into a stratified retina, similar to a normal one, through differentiation. RPE cells that remain at the original site renew the RPE itself. Some key TFs and signaling pathways that are regulators of retinal regeneration in frogs were investigated. In particular, FGF2 was shown to accelerate RPE transdifferentiation in vitro and in vivo and is necessary to maintain the Pax6 expression and cell proliferation at a certain level [21,45]. The expression of the *rx* gene is necessary for the formation of the NR regenerate after the partial incision of the retina in pre-metamorphic *X. laevis* [46,47]. The *rx* knockdown could impair regeneration by leading to a drastic reduction in proliferation [47]. Like Urodela, Anura possess CMZ. The latter contains a heterogeneous pool of poorly differentiated cells with regional differences in genetic expression. These differences reflect the vector of advancement of poorly differentiated progenitor cells by the retinal differentiation pathway from the periphery to the center of the eye. The marker genes of stem cells, *pax6* and *six3*, have a high level of expression on the extreme periphery of the retina; more distally located cells (towards the center) express genes that are markers of pro-neural differentiation: *delta*, *notch*, and *neuroD* [48]. Cells of the most distal CMZ region directly contribute to retinal regeneration in Anura [43]. The expression of *Rax* genes was also detected in the CMZ of the *X. laevis* mature retina, which additionally indicates the progenitor properties of these cells [49]. The expression of proto-oncogenes, the cell cycle regulators *c-myc* and *n-myc*, was recorded from the CMZ of Anura [50]. Cases of the regeneration of NR only due to CMZ cells were also described in Anura. In the study by [43], the authors used *Xenopus tropicalis*, a frog species that exhibits CMZ-derived regenerated retina. In this model, RPE cells migrated towards the eye cavity and the RVM, but the transdifferentiation of these cells was not evident. However, a complete retina regenerates approximately 30 days after the whole retinal removal. These data are a vivid demonstration of the probability of differences in the proportion of involvement of RPE and/or CMZ cells in retinal regeneration, not only with different modes of injury discussed below, but also when using different species from the same class of amphibians in the experiment.

In studies of issues related to retinal regeneration in caudate and tailless amphibians, Müller glia cells still remain in the shadow. The response of the Müller cells of the Urodela and Anura retina to damaging effects has not been systematically considered. According to our data [51] and other observations [23], it is preliminarily assumed that, in adult representatives of both amphibian groups, Müller cells are in a quiescent post-mitotic state, while showing the capability of proliferation and gliotic response. When this population was examined in semithin sections through the damaged retina of newts (*P. waltl*) and using the marker ^3^H-TdR, its incorporation was detected in Müller cells, which indicated the DNA synthesis [52,53]. It is also known that the Müller cells of caudate amphibians are capable of gliotic response to changes occurring in the retina. This typical, well-known reaction to retinal damage [54] was observed in newts with NR detachment and with variations in intraocular pressure caused by simulated microgravity conditions [51,55]. Müller cells’ gliotic response was also found in tailless amphibians [56]. As was noted in the study [56], Müller glia cells in *X. laevis* respond to retinal damage with morphological changes characteristic of gliotic response: cell hypertrophy, an increase in the number of cells, and the formation of gliotic scar. Additionally, the study [57] used a mechanical needle poke injury on model transgenic *X. laevis* tadpoles, allowing for conditional photoreceptor cell ablation, showing that Müller cells are capable of the proliferation and even replacement of lost cells following damage/degeneration in the retina. Thus, it was noted that the extent of cell cycle re-entry appeared to be dependent on the age of the animal, with a refractory period in early tadpole stages [57]. Undoubtedly, the issue of the involvement of Müller cells and their contribution to retinal regeneration in amphibians of various species and different ages requires further elucidation.

## 3. Retinal Damage, Methods, and Consequences

In this section, we consider the known methods for retinal damage/regeneration in amphibians in order to show how different the regenerative responses of regeneration source cells can be, and how they depend on the damage method. In early [25,58] and later [28] experiments, the microsurgical removal of NR through an incision on the dorsal side of the eye along the limbus, while preserving the RPE and a portion of CMZ cells tightly adjacent to the RPE at the eye periphery, was used to study retinal regeneration in caudate amphibians. Another approach to damage in Urodela was cutting the optic nerve and the surrounding blood vessels [59,60]. This manipulation leads to the retrograde death of neurons and, as a result, the degradation of the original NR. Then, this is followed by a complete restoration of NR due to RPE and CMZ cells. The third method, the separation of RPE and NR by microsurgical detachment of NR, was used in the experiments on newts [53,61] and also birds [62]. The detachment of NR from RPE has its advantages because it allows the observation of the changes occurring in response to the injury simultaneously in RPE and NR. In rare cases, we used a “mild” method of retinal damage: the irradiation of the eyes of mature Urodela with bright light [63]. Cells’ responses to various damaging procedures had similarities and differences. The similarity was in the mandatory activation of RPE and CMZ cells. The NR detachment and irradiation of Urodela eyes with bright light made it possible to additionally observe the regenerative responses of bipolar-like cells, another resource identified but not yet sufficiently studied [41]. 

The differences between various types of operations were related, first, with the size of the contribution of RPE and CMZ populations to the NR formation de novo [64]. After the surgical removal of NR, RPE cells made the major contribution to regeneration; after cutting the optic nerve and blood vessels, CMZ cells provided the greatest contribution. A similar response with the dominance in CMZ cell regeneration was also observed in retinal detachment in newts against the background of RPE cell activation. The least pronounced responses from RPE and CMZ occurred in the case of the irradiation of the retina with bright light that did not cause significant cell death under experimental conditions [63]. 

The retina of tailless amphibians was also subjected to various types of injury to induce regeneration at the larval and tadpole stages. The article [20] presents the results obtained at the post-metamorphic, mature *X. laevis*. In this study, NR was removed with the preservation of the RPE and RVM layer, capillary-rich basal membrane, bounding the inner margin of the retina. As already mentioned, upon retinectomy, Pax6-positive pigmented RPE cells that detached from Bruch’s membrane migrated towards the RVM. At the RVM, RPE-derived proliferating cells undergo transdifferentiation, become stem-cell-like, and form a neuroepithelium [20]—as reviewed in [23]. RPE cultivation within the tissues of the posterior wall of the eye and the layer of RPE cells isolated from the underlying tissues was also used [20,65,66,67,68,69]. 

Unlike the genome of caudate amphibians, the genome of *Xenopus* is well annotated [70,71], which allows one to conduct studies on the level of RNA sequencing and proteomics using this animal model [72,73]. There are examples of the use of the chemogenetic selective removal of photoreceptor cells in *X. laevis* for the purpose of targeted damage of NR [74,75]. Martínez-DeLuna and Zuber [75] used the XOPNTR transgenic line in which the *Xenopus* Rhodopsin promoter drove the rod photoreceptor-specific expression of the bacterial enzyme, NTR. The exposure of transgenic tadpoles to the antibiotic Metronidazole (Mtz) for 2 days completely ablated rods by day 7 after the initial Mtz exposure. The removal of Mtz allowed rods to regenerate and made rod-specific ablation amenable for various regeneration studies. Earlier, in the study [76], the authors developed a transgenic *X. laevis* model for retinitis pigmentosa where rod apoptosis was initiated by the administration of a normally innocuous compound AP20187. Transgenic *X. laevis* expressed a modified caspase-9 (iCasp9) under the control of the *X. laevis* rod opsin promoter. These experimental models differ from other inducible models (such as light damage) in that the cell death pathway is well characterized, and rod photoreceptors are the only cells affected by the primary insult. In their study, the authors in [76] noted that, in addition to growth from the retinal periphery (CMZ), cell bodies immunopositive for anti-rhodopsin occurred in the central retina 2 days after AP20187 administration. After 5 months, the regeneration of photoreceptor phenotype cells was detected, presumably due to replacement by progenitor cells localized in the inner retina against the background of the death of pre-existing rods. These cells were assumed to be similar to those described earlier after NR damage in *X. laevis* larvae [65] and in mature newts after retinal detachment [40]. It is worth noting here that a wide variety of techniques to damage eyes of various animal classes for the induction of retinal regeneration and the modeling of retinal diseases has recently been presented in a detailed review [77]. 

Injury may affect the specific strategies of behaviors of cells involved in retinal regeneration. These differences in different types of damage may be assumed to be caused by intercellular and inter-tissue signaling, which is created both as a result of altered cell-to-cell relationships, cell death and its scale, and due to a number of other factors, e.g., physical such as variations in intraocular pressure, the tension of RPE and NR, etc. Signaling factors, in turn, are part of the regulatory niches of source cells of NR recovery, which set up the behavioral pattern for them: activation, migration, entry into the proliferative phase, and the onset of conversion and differentiation by a new pathway. Their further fate is determined by the mechanisms of self-organization during the in situ and off-site morphogenesis of retina regenerate [78,79]. However, as shown using the model of retinal regeneration in another animal class, zebrafish, damage by ultraviolet, bright or flashing light, chemical agents, or mechanical injury can initially trigger similar mechanisms for maintaining cell viability [80,81,82]. The known methods for damage to the amphibian retina (Urodela and Anura) are presented in Table 1.

## 4. Early Events That Occur after Separation of Neural Retina and Retinal Pigment Epithelium

### 4.1. Cell Stress and Cell Death after Retinal Damage in Amphibians

The maintenance of general cellular metabolism and redox homeostasis plays a major role in the control of the condition of RPE cells [85,86]. There is both conservatism and variability in relation to the molecular participants of signaling pathways associated with maintaining homeostasis and providing endogenous cell protection [87,88,89]. The phylogenetic variability of the redox-sensitive TFs of NRF families has been found, which may have adaptive significance, as well as determine the differences in cellular and regenerative responses with the destruction of the interactive cooperation of RPE cells and retinal photoreceptors [90].

Early events after the separation of NR and RPE in vertebrates are associated with disorders of redox homeostasis, balance between prooxidant and antioxidant systems, and the down-regulation of molecules that are components of the visual cycle [91,92,93]. Studies of the *X. laevis* tadpole retina based on proteomic analysis showed that the removal of RPE not only disrupts outer segment assembly but also alters the protein expression profiles of photoreceptors [94]. The intensification of free radical oxidation processes, along with inflammatory factors, activates the operation of regulatory and enzyme systems aimed at protecting cells from OS [95]. However, the data obtained on different damage models indicate that OS is utilized as part of the protective mechanism of the general regulatory system in the initiation of cellular processes aimed at tissue repair [96,97,98,99,100]. There is information about certain components that make up protective systems when initiating retinal regeneration in amphibians. In both caudate and acaudate amphibians, heat shock proteins (HSPs 70, 90) were identified [101,102]. Experiments using PCR, Western blot hybridization, and immunohistochemistry showed that, in the tissues of the non-operated *P. waltl* eye, including RPE and CMZ, the proteins HSP70 and HSP90 and th et transcripts of the corresponding genes are constitutively expressed at a low level [102]. They were assumed to be involved in the general mechanism of maintaining cellular viability (due to their better tolerance to cellular stress) after the separation of RPE and NR [103]. It is worth noting the major functions of chaperone proteins are to prevent the intracellular accumulation of cytotoxic proteins and regulate protein folding [104]. As regards RPE cells, chaperones can prevent RPE cell apoptosis by being involved in the activation of phosphorylation reactions in the PI3K/Akt signaling pathway, which ensures RPE cell tolerance to OS. HSP70 is involved in the regulation of autophagy-mediated protein proteolysis in the RPE [105]. In mammals with the experimentally induced shutdown of chaperone proteins in RPE cells, reactive oxygen species’ (ROS) accumulation occurred under OS conditions, followed by retinal cell degeneration [106,107].

Cell death accompanies and is largely a cause of all regenerative processes without exception. Apoptosis, programmed cell death that may reshape remaining tissues and recycle resources, is required for tissue regeneration in animals of a variety of classes [108,109,110,111,112]. Cell death is known to accompany the main stages of self-organization of the retina during its histogenesis in development [113,114,115] and the morphogenesis of retinal regenerate during regeneration in adult animals [116]. Retinal detachment results in the early activation of stress-response-specific signaling pathways. Among the different types of cell death, apoptosis is the major one, whose basic features and morphological changes inherent in all cells are well described. The mechanism of apoptosis induction is not unique and includes different pathways that depend on the cell type and tissue damage [117]. The key OS-dependent signaling pathways of apoptosis in RPE are mediated by JNK/SAPK (c-Jun N-terminal kinase/stress activated protein kinase), p38, ASK1 (apoptosis signal-regulating kinase 1), and PKC (protein kinase C) [118,119]. The multistep cascades of caspase-dependent and caspase-independent signaling pathways mediate apoptosis [117], with the prevalence of one or the other depending on the intensity of cellular stress. The main molecular messengers responsible for the proapoptotic effect are the AP-1 complex, regulatory kinases, and effector caspase 3 [91]. This adaptive response may enable the RPE and photoreceptor cells to survive the acute phase of retinal detachment [120].

Ferroptosis, autophagy, and necroptosis, also occurring in OS conditions and described for mammalian RPE cells, are considered in addition to apoptosis [117,121,122]. The RPE metabolism is closely related to the functioning of the neighboring tissues (photoreceptors, Bruch’s membrane, and choroidal coat). Intercellular signaling cascades coordinating the work of the visual cycle of biochemical machinery within RPE and photoreceptors are assumed to be decisive for maintaining the cells’ viability and differentiation. Separation between RPE and NR leads to a visual cycle suppression and an excessive accumulation of photochemically active molecules, ROS, and their metabolic products. This can trigger the RPE cells’ and photoreceptors’ apoptosis. The key components in apoptotic signaling pathways are already known [91,123]. Despite this fact, the contribution of apoptosis to retinal regeneration is not fully understood. It is likely that the apoptosis of retinal cells does not directly lead to cell death, but occurs as a secondary event following the previous chain of biochemical reactions which immediately follow the RPE↔NR separation [124].

As regards the Urodela RPE, after the separation from NR in the case of the retina’s removal or detachment, no mass cell death is observed in the RPE layer. Only a very small number of cells in the state of apoptosis are found morphologically and using the TUNEL method, mainly in the central area of the fundus of the eye in newts (personal observations). However, the remaining RPE cells are mainly activated, dedifferentiated, proliferated, and converted in this region, subsequently becoming a source of NR repair. Another study, based on labeling by the TUNEL method, described the presence of apoptotic cells in retinal regenerate cells in situ after the surgical removal of the original NR in newt [116]. In this study, no information was provided about cell death in the posterior sector of the eye shortly after the removal of NR. The first wave of apoptosis was observed at the stage of entry of NR anlage cells into the active proliferative phase; the second was observed during the histogenesis of the formed de novo NR. Dying cells were found in small numbers among CMZ cells that persisted after surgery [116]. The data on NR regeneration in Urodela, considered together, suggest that the cells of Urodela RPE exhibit a relatively high viability, which allows them, escaping death, to serve as a source of NR regeneration. This is indirectly confirmed in the observations [64,82] made during an in vitro cultivation of the posterior sector of the eye after NR removal. The high viability of Urodela RPE cells can be explained by assuming the resistance to factors inducing cell death and/or by an intrinsic system of protection of these cells from cellular stress [90], as well as by the low rate of cellular metabolism in these animals. The low metabolic expenditure may allow many amphibians to survive and overcome the effects of negative factors including hypoxia and oxidative stress [125,126,127].

The fate of the RPE cells of *X. laevis* frogs was observed during RPE explantation, cell dissociation, and cultivation in a low-density suspension culture with the lack of proliferation [128]. The cell viability in these cultures was essentially undiminished over the initial 2 days. This means that, as in the newt, no rapid death of RPE cells occurs after damage in vivo and under the conditions of isolation and cultivation in vitro. In the same study [128], it was additionally found that the exposure to fibronectin, collagen IV, laminin, as well as insulin added to the medium contributes to the longer viability of *X. laevis* RPE in vitro. Plating cells on a fibronectin-coated substratum significantly enhanced their survival rate: the number of cells counted as alive at 1 week was 80–90% of the initial level [128]. Earlier, it was found [129] that, in primary cultures, *X. laevis* RPE cells retain all the phenotypic characteristics of the epithelium in situ for 7–10 days (observation time), as well as the ability to establish de novo adhesive interactions with the autologous retina as early as at 3 h after co-cultivation. There is a report [130] that the p75 neurotrophin receptor (p75NTR), a member of the tumor necrosis factor receptor family, is involved in the promotion of survival and is a factor of cell viability in the developing *X. laevis* retina. In this case, the lack of dependence of p75NTR functioning on the developmental context is assumed [130]. This fact is noted to contrast with the reported role of p75 (NTR) in developing the retinae of other species, i.e., its expression is a feature of Anura [130]. Thus, damage to amphibian RPE cells after injury, and separation from NR can be assessed not only in the context of subsequent apoptosis as such, but rather as an event activating the mechanisms of viability and inducing regenerative responses.

Recently, discussions in the literature have been concerned with the issue of the release of the so-called “apoptotic cell-derived extracellular vesicles” (ApoEVs) by dying cells [108,131,132,133]. Their circulation and transmission to neighboring and also distantly located cells is assumed. ApoEVs are considered a tool for efficient intercellular communication. ApoEVs are a kind of container for transferring nucleic acids (including microRNAs), proteins, and lipids, signaling and stimulating the regenerative responses of target cells [108]. It is also assumed that the release of ApoEVs accompanies the early phase of cell death and serves to regulate immune responses from residential and circulating inflammatory cells, as well as to control the cleaning of tissue from aging cells. We set up experiments on the ARPE-19 line of the human eye RPE cells in vitro exposed to media conditioned by the newt retina cultured in vitro. The results of this study [134] provided interesting information that, nevertheless, requires further study and analysis. Preliminary short-term exposure to the medium conditioned by newt retina destabilized the epithelial differentiation of ARPE-19 cells and induced the manifestation of molecular traits characteristic of the early neural differentiation stages.

With the use of explanted hind limb cells (A1 cells) in the newt *Notophthalmus viridescens*, the production of extracellular vesicles like EVs by A1 cells after their isolation was first detected [135]. The medium conditioned by A1 cells contributed to the protection of neonatal rat cardiomyocytes from apoptosis caused by OS in vitro. A treatment of newt cells with an EVs biogenesis inhibitor reduced the EVs output and attenuated the conditioned media’s protective effect [135]. Extracellular EVs surrounded by a lipid bilayer membrane, which were produced by newt A1 cells, were nanoparticles (100–150 nm) and resembled mammalian exosomes. The difference was in significantly higher contents of RNA (among which mRNAs were recorded) and proteins encoding nuclear receptors, membrane ligands, and TFs [135]. The aforementioned study suggests the probability of the production of similar EVs by cells damaged or entering apoptosis after surgery in other regeneration models, including cases of retinal regeneration in Urodela.

EVs were also isolated and analyzed from developing Anura [136]. The study used a model of the early developmental stages in *X. laevis* accompanied by extensive cell death. The authors managed to collect rich EV material from extracellular spaces, which served for biochemical, transcriptomics, and proteomic studies. The study, in addition to the presence of the EVs themselves, demonstrated the probability of their trafficking and absorption by neighboring cells. It was emphasized that the detection of EVs in very distant but regeneration-competent animals such as amphibians and the sea cucumber *Apostichopus japonicus* supports the idea that EV-mediated communication is conserved across many animal models [136,137]. Thus, these data, despite remaining very scarce, can be preliminary regarded as evidence for the role of EVs as one of the possible mechanisms of cell–cell communication, regulating the pro-regenerative activity of the cells of the posterior wall of the eye in Urodela and Anura shortly after injury and under the action of apoptotic signals.

### 4.2. Disturbance of Retinal Cell Contacts, Rearrangement of Cytoskeleton and ECM

The operation-induced destabilization of cell differentiation and homeostasis in the Sclera + Choroid + RPE↔NR system initially suggests changes in cell-to-cell contacts and ECM. In amphibians, as in other vertebrates, RPE cells are normally in an adhesive connection with photoreceptors and in close connection with each other and with the Bruch membrane (BM) lining the RPE, which includes the cytoplasmic membranes of RPE cells from the basal side [138]. The main components of the BM are as follows: collagen IV, laminin, entactin, heparan sulfate, and proteoglycan 2, which play a crucial role in vascular integrity [139]. RPE cells, normally performing a boundary function, lose their normal connection with their entire 3D environment after damaging surgery. The cytoskeleton of RPE cells is a reflection of its epithelial and functional specialization, as well as phenotypic plasticity. The state of the cytoskeleton of RPE cells in the newt *P. waltl* under normal conditions and at the onset of conversion was previously studied. The removal of NR, as well as the detachment of NR, leads to the inhibition of the expression of intermediate cytoskeletal proteins: cytokeratins in cells in the RPE layer. The same is immediately observed after the isolation and dissociation of RPE cells taken from a non-operated newt eye [140,141]. However, the expression of pan-neural proteins NF-200 is initiated in the first RPE-derived cells that still retain pigment granules in the cytoplasm and express RPE65, but are displaced and leave the layer [142]. The high rate of switching to reading in genes that encode intermediate filament proteins NF-200 in the absence of proliferative activity indicates the special state of the system of expression which regulates specific differentiation genes in sexually mature Urodela, which is permissive for the rapid conversion of RPE cells. According to some assumptions [143], the conversion of RPE Urodela is provided by the expression of pioneer TFs and the demethylation of regulatory elements of photoreceptor genes. It is also known that the microtubule reorganization can influence the shapes of RPE cells [144]. The exit of RPE cells is accompanied by a change in their shape from cuboidal to oval and a variation in the nucleus-to-cytoplasm ratio. It was shown [145] that, at this stage, as well as at later stages, the formation of the NR regenerate in the newt occurs in the context of the expression of stathmin. Stathmin is a small cytoplasmic phosphoprotein known to be a microtubule regulator involved in cell cycle regulation [146].

The issue of NR regeneration-initiating changes in the cell-to-cell contact system in Urodela (newt *Cynops pyrrhogaster*) was addressed by Yasumuro et al. [31]. The authors managed to record the nuclear translocation of β-catenin in RPE cells induced by the attenuation of cell–cell contact. This phenomenon could also be observed in the case of the incision of RPE or its treatment with ethylene glycol tetraacetic acid (EGTA), a Ca^2+^ chelator that disrupts cadherin-mediated cell–cell adhesion. The ongoing translocation of β-catenin in RPE with the simultaneous decrease in the immunoreactivity of N-cadherin (N-Cad), according to [31], can facilitate the entry of cells into the S-phase. Preliminary data from an immunochemical study of the localization of the zonula occludens (ZO) proteins, associated with tight RPE contacts, showed a redistribution of ZO-1 over the cell surface shortly after NR detachment in newts, *P. waltl* [147]. A decrease in ZO-1 expression was detected at an early stage of retinal regeneration in the newts *Cynops pyrrhogaster* at the level of transcriptomic analysis [148]. There are also reports that the RPE-derived progenitors of the retina anlage are capable of re-establishing intercellular interaction after entering the proliferative phase by connecting with each other via gap junctions. This suggests the presence of their cytoplasmic communication during an early phase of NR regeneration [149]. These few studies together have revealed a wide range of events affecting the cytoskeleton, cell surface, and intercellular contacts in cases of the disruption of the 3D topological connections of RPE cells with their cellular environment.

The ECM plays an essential role in the stabilization of the cell phenotype and, vice versa, in the manifestation of its plasticity properties [150,151]. In the case of NR detachment in newts, we previously noted dynamic changes in the localization and expression of fibronectin (FN), a glycoprotein of ECM. After surgery, the immunoreaction of FN antigens for the treatment with appropriate antibodies decreases in the basal surface facing the Bruch’s membrane and is redistributed to the lateral RPE cell surfaces [152,153]. FN is an adhesive ECM component of the Bruch’s membrane and promotes the attachment of RPE cells to BM. A decrease in its content and redistribution can be considered one of the mechanisms that allows the exit of RPE cells from the layer and displacement towards the eye cavity. Other ECM components also tend to be regulators of plasticity and the subsequent conversion of newt RPE cells by the neuronal and glial pathway: tenascin, laminin and N-CAM [154], heparan sulfate proteoglycan, and nidogen/entactin [153]. Active ECM remodeling was also recorded during the lens regeneration in Urodela, a model which is similar in the pattern of the process to the NR regeneration in the same animals. The studies [155,156] showed the involvement of ECM components in the conversion of pigmented iris epithelial cells. In the site of lens regeneration, source cells in the pupillary margin of the dorsal iris, the activity of hyaluronidase proved to be 1.7- to 2-fold higher than that in the ventral iris prior to lensectomy and later, when lens regeneration progressed. It is suggested that the high level of hyaluronidase activity in the dorsal iris may promote the turnover and remodeling of ECM components required for iris cell-type conversion [155]. In [156], the expression of 373 genes in the iris of an adult newt was analyzed before and after lensectomy. In particular, several of these genes were found to be involved in tissue remodeling, such as matrix metalloproteases *MMP9*, *TIMP*, and genes coding collagenase *and* cathepsin. The function of the metalloproteases is activated by ROS and is associated with the degradation of ECM proteins in the BM. Changes in ECM, along with OS, increase the permeability of the blood–tissue barrier and promote the extravasation of inflammatory factors in the Sclera + Choroid + BM + RPE ↔ NR system [157]. Persistently high MMPs levels may act to prevent excessive new ECM production as the damaged resident tissue undergoes histolysis and is prepared for the deposition of the regenerative matrix in the case of injury in caudate amphibians [158]. Available data show that the composition of ECM, produced during cell response, may also have an anti-fibrotic effect that helps recapitulate embryonic development and promote a regenerative response [158,159]. It is likely that, in a more general sense, the results of those studies might also be applicable to retinal regeneration in Urodela.

The models for the ex vivo cultivation of posterior wall tissues of the Xenopus eye showed that the retention of the connection of RPE with the underlying tissues keeps it from losing its original properties, cell differentiation, and conversion. Conversely, the separation of the retina provokes RPE to reprogram. A study of key TFs in the retinal regeneration system in *X. laevis* showed the upregulation of the *Pax6* and *Rax* genes during RPE cell conversion [160]. Transgenic *X. laevis* were obtained which had an EGFP expression occurring under the control of the *Rax* promoter. The expression of *Rax-EGFP* was analyzed during NR regeneration in a tissue culture model. The *Rax-EGFP* expression was shown to precede the *Pax6* expression, and the expression of both genes occurring in RPE cells lost contact with the BM rich in ECM components. It was emphasized that these temporal changes in cell–cell and cell–ECM interactions regulate RPE transdifferentiation and retinal regeneration.

Previously, the role of ECM components in the differentiation state of RPE cells in *Rana* frogs was demonstrated using in vitro conditions [67]. It was found that laminin within the cultivation substrate profoundly influenced RPE cells transdifferentiating into neurons. The use of antibodies against laminin–heparan sulfate proteoglycan led to the inhibition of NR regeneration in *Rana* [68]. It was assumed that retinal regeneration is initiated by changes in the ECM composition which regulates RPE cells’ contacts early in the process [67]. The presence of a high concentration of laminin in the RVM involved in the formation of NR regenerate in frogs indirectly confirms this assumption [20]. The role of MMPs was investigated using the model for the transdifferentiation of *X. laevis* RPE cells under conditions of in vitro tissue cultivation [161]. It was found that, after the NR removal, there was an upregulation of MMPs (Xmmp9 and Xmmp18) in the surrounding RPE cells and choroid at the stage of the displacement of RPE cells from the layer. A potent MMP inhibitor, 1,10-PNTL, suppressed the RPE cell migration as well as the further proliferation and formation of a retina-like structure in vitro. The authors of the study associated the role of MMPs with the expression of the inflammatory cytokine genes IL-1β and the factor TNF-α that exhibited the upregulation of MMPs. When the inflammation inhibitors dexamethasone or Withaferin A were applied in vitro, RPE cell migration was severely affected, suppressing RPE transdifferentiation. The article emphasizes the important trigger role of *X. laevis* proinflammatory cytokines in ECM reorganization during NR regeneration in amphibians.

Thus, the study of the role of ECM using models of NR regeneration in caudate and acaudate amphibians gives reason to assume the significant importance of ECM remodeling, the differential expression of its components, particularly MMPs, and their relationship with proinflammatory responses at the initiating stages of the NR regeneration process in amphibians. The unique molecular composition of ECM suggests that the newly synthesized ECM may antagonize fibrosis and promote regeneration in caudate amphibians.

### 4.3. Role of Immune System in NR Regeneration in Amphibians

The activation of the immune system after injury is a prerequisite for healing and regeneration [162]. Tissue damage first stimulates the recruitment and activation of cells of the immune and circulatory systems, which, in turn, produce a number of regulatory factors and are capable of mobilizing source cells and triggering the mechanisms for the regeneration of tissue damage. The alleged role of the immune system in ECM remodeling is mentioned above. However, the immune system is also, and above all, the first line of defense and a participant in the reparative processes in both epimorphic regeneration and scarring [163,164,165]. The responses of the immune system primarily depend on its own characteristics that are species- and age-specific for animals, as well as on the characteristics of the injury inflicted. An inverse relationship is assumed between the maturity of the immune system and the ability to regenerate [166]. Amphibians belong to the category of animals having a “primitive” immune system, which, according to [164,167], is one of the explanations for their high regenerative ability. The ancient and primitive characteristics of the immune system in amphibians determine its ability not only to regulate but also to be directly involved in tissue regeneration [168,169]. The specifics of immune responses in caudate amphibians and their long-term healing/regeneration without visible inflammation and scarring have been previously reported [170,171].

The relationship between the immune system and regeneration processes in various tissues and organs of salamanders was observed in detail in the review by Bolaños-Castro et al. [169]. According to this generalized information, the innate and adaptive components of the immune system in Urodela play a role at critical stages of the regeneration of many tissues, from the moment of damage to the morphogenesis of regenerates. In a recent study [172], the structures, functions, and features of the lymphatic system in Urodela were considered in detail. This study gave reason to assume significant anatomical differences between urodeles, anurans, and mammals, which should be taken into account when addressing the issues related to their unmatched regenerative ability. In particular, no significant changes in regeneration were found after the excision of major components of the newt lymphatic system vasculature. These data suggest that the adaptive immune response may not be essential for regeneration but, instead, may have a regulatory role [172].

The process of inflammation, following the initial injury signals, unifies the cellular post-injury response in animals. The damage of proteins, ultimately resulting from cell lysing, and the release of molecules such as ROS as a result of mitochondria overproduction, promote the synthesis of pro-inflammatory cytokines [173]. ROS and cytokines serve as primary signals for many components of cells’ stress response [174]. ROS are necessary for the recruitment of immune cells to the wound site, as shown using the model for the damage and regeneration of the caudal fin in zebrafish larvae at 3 days post-fertilization [175]. Besides hemostasis, the sequential influx and action of immune cells—neutrophils, macrophages and lymphocytes—also constitutes the inflammatory phase of wound repair. The emergence of immune cells at the site of injury is one of the first signs of initiation of the recovery process. Here, the immune system cells demonstrate multiple roles, taking part in the removal of debris, the cleaning of tissue from dying cells, and the production of a number of signaling molecules that, in turn, induce subsequent responses, particularly from the source cells of regeneration [164,176,177]. As described above, in the case of the regeneration of mature Urodela, such events include the entry of RPE cells into the mitotic phase and the onset of the dedifferentiation and reprogramming into neural cells. The responses of amphibian immune cells and what regulatory molecules are produced in this case are poorly known to date. Recently, a study in this field was conducted using a model for limb regeneration in *Ambystoma* [178]. The authors used the enriched fractions of peripheral immune cells and single-cell sequencing to identify the genetic markers of erythrocytes, neutrophils, macrophages, and lymphocytes. Since immune cells are known to infiltrate areas of injury, samples were also taken from the area of a regenerating limb. Under normal conditions and during regeneration in *Ambystoma*, heterogeneous cell populations were identified where circulating monocytes and resident macrophages were present. These cells were assumed to have specific functions and the property of responding in different ways to activating stimuli [178].

Previously, studies using the model for heart [179] and limb [176] regeneration in *Ambystoma* revealed the role of macrophages in the cleaning of the damage site from dying and aging cells and in the production of pro- and anti-inflammatory molecules involved in ECM remodeling. The essential role of macrophages was emphasized by the fact that the depletion of these cells through the injection of clodronate-encapsulated liposomes blocked limb regeneration [176]. We should note here that, according to the observations [176], neutrophils and macrophages can already be detected on the first day after limb amputation. In view of the specifics of energy metabolism and the generally low metabolic rate in Urodela [125,126], this indicates that these cell populations of the Urodela immune system are rapidly recruited and are among the initiators of regeneration. In studies of skin regeneration in terrestrial axolotls (salamanders), these animals exhibited a reduced hemostatic response, increased leukocyte infiltration, and a higher total leukocyte numbers. Furthermore, lower neutrophil levels, similar durations of inflammation, faster times to complete re-epithelialization, a delay in new transitional matrix production, and differences in the relative composition of the new ECM were recorded. These processes contribute to scar-free healing in caudate amphibians compared to mammalian wound repair. It was assumed that the reduction in neutrophil levels occurs through passive aggregation or chemotaxis, and that their loss may increase the re-epithelialization rate [158]. For mammals, the emergence of macrophages as an important source of pro- and anti-inflammatory signals after damage was only recorded at 48–96 h post-injury [180]. It is also important to note that, in contrast to mammals, Urodela have a highly efficient mechanism of senescence immune surveillance, preventing not only their accumulation, but also the de novo emergence [181]. During limb regeneration in salamanders, the recurrent emergence of aging cells occurs in the context of the formation and expansion of progenitor cells. In this case, aging cells are rapidly eliminated by macrophages [181]. Degenerative changes in the outer retina were recorded from the retina of aging tiger salamanders (*Ambystoma tigrinum*): the loss and disruption of outer segments, RPE abnormalities, and cell death. It was noted that, in this context, the number of macrophages and lymphocytes within the retina significantly increased [182]. This is further evidence of the broad and direct role of immune responses, e.g., the mobilization and active involvement of macrophages, as triggers of regeneration after damage.

With the exception of the latter, the above examples (limb and heart) do not concern the brain and retina that are known to be protected by the blood–brain barrier and the blood–retina barrier, respectively [183,184]. Taking into account the existence of the blood–retina barrier, the involvement of newly arriving inflammatory cells in addition to existing resident ones in retinal regeneration in Urodela seems interesting and unusual.

In earlier studies of newts, the role of macrophages of different origins in NR regeneration was noted. With the initiation of retinal regeneration in Urodela, cells localized on both sides of the RPE, morphologically identical to macrophages, are soon detected after the separation of RPE and NR (Figure 2).

Among them, there may be both newly arrived macrophages and residential ones. In an experiment with the simultaneous removal of NR and lens in mature newts, the dynamics of migration and proliferation of macrophages was observed [185]. After the pulsed administration of [^3^H]-TdR, labeled macrophages (0.2–5.9%) were detected in the eye cavity as early as at 2 days post-operation (dpo); at 12 dpo, the proportion of the macrophages in the DNA synthesis phase reached 73%. It is assumed that non-dividing macrophages arrive in the site of damage where they and resident macrophages show the ability to actively reproduce.

The accurate identification of the origin of monocytes/macrophages is also complicated by another noteworthy fact that is very characteristic of Urodela, which was noted many years ago. A study of this process of retinal regeneration in newts, in addition to the displacement, dedifferentiation, and conversion of RPE cells followed by the formation of NR regenerate, revealed another small population of cells. These cells were displaced from RPE and changed their epithelial pigmented phenotype to a phenotype bearing the morphological characteristics of macrophages [60,186]. This observation was made using electron microscopy in the course of NR regeneration after retinal removal or optic nerve cutting in the newt. It was found that, after NR removal in Urodela, some RPE cells leave the layer, move in the vitreal direction, and phagocytose retinal cell remnants rich in melanin granules. Due to this ability, they were referred to as “melanophages” [60,186]. This ability of “melanophages” serves the dedifferentiation of RPE cells, freeing them from their main specific trait: melanin granules in cellular debris. This ability is probably initially due to the phagocytic ability of RPE cells, which normally serves the performance of a visual function, including the absorption and digestion of the renewing outer segments of photoreceptor cells [187]. Thus, the data obtained using Urodela demonstrate the multiple active roles of macrophages in the initiation and progress of regeneration. Of particular note are their different origins and localizations in the posterior sector and the cavity of the eye, which is the site of RPE cell reprogramming immediately after damage.

Recently, a broad study was conducted on the role of macrophages in the regeneration of the lens that occurs through the conversion of iris cells in the newts *N. viridescens* and *P. waltl* [188]. The authors of the study generated a transgenic newt reporter line, in which macrophages can be visualized in vivo. This tool made it possible to identify the exact localization of the macrophages during the process. In parallel, there was a study of changes in genetic expression using the bulk RNA-seq to uncover the upregulated expression of genes linked to macrophages/monocytes, inflammation, and other processes. The technique of suppressing macrophage activity using clodronate liposomes was also employed, which made it possible to observe the inhibition of de novo lens formation in both newt species. Moreover, the macrophage depletion induced the formation of scar-like tissue, an increased and sustained inflammatory response, an early decrease in iris pigment epithelial cell proliferation, and a late increase in apoptosis. However, a secondary injury alone or the addition of FGF2 restarted the scar resolution and the regeneration processes. The findings in the study indicate the importance and multiple functions of macrophages in facilitating a pro-regenerative environment in the newt eye, helping to resolve fibrosis, modulating the overall inflammatory landscape, and maintaining the proper balance of early proliferation and late apoptosis between scarring and regenerative phenotypes [188].

Thus, the multifunctional and important role of macrophages noted in this section seems to be a fundamental feature of Urodela: a set of mechanisms initiating tissue regeneration. Further research should apparently be aimed at collecting information about the contribution of tissue and circulating macrophages and the production of specific molecules included in the system, regulating the conversion of cell sources of NR regeneration by these cells. It is also possible that the molecules above mentioned, released by damaged tissue, can be sensed by tissue-resident macrophages, which then secrete chemoattractants and pro-inflammatory cytokines to recruit circulating macrophages and other cells of immune response in Urodela.

It has long been noted that, within the Amphibia class, significant differences exist between the immune systems of Urodela and Anura [189]. Urodela, compared to Anura, are not only more simply organized but also have a certain “immunodeficiency”—a weakly expressed immune response and a weak immune memory and cellular immunity [189]. In Anura, the regenerative ability correlates with the course of ontogenetic development, including the course of the adaptive immune system formation [190,191]. The success of tissue regeneration in Anura is determined, on the one hand, by a set of interactions between cell sources of regeneration and, on the other, by cells and factors of immune system [191].

Unlike Urodela, the role of macrophages in Anura was shown to inhibit limb regeneration at the stages prior to metamorphosis [192]. Moreover, a hypothesis was advanced that the evolution of the adaptive immunity in tetrapods while efficiently preserving adult self-condition determined the loss of tissue regeneration since the embryonic antigens evocated in blastema cells are removed by the immune cells of the adult [192]. It is likely that the immune surveillance that occurs in amphibians during regeneration is taxon-specific and depends on the developmental stages and age: this stimulates the regeneration in the case of Urodela and inhibits it in Anura tadpoles [170]. It is also known [193] that in *Xenopus* tadpoles, the crush lesion of the optic nerve leads not only to the rapid degeneration of its axons, but also to their subsequent regeneration within only five days in the context of an abundance of debris and dead cells. Thus, an immunohistological analysis with the monoclonal antibody 5F4 showed a rapid and extensive microglial/macrophage response to the site of ON crush. Macrophages were observed in the nerve at the site of the lesion within 1 h, and the response peaks were within 3–5 days, shortly before axonal regeneration was initiated [193]. Thus, the pattern of relationships between immune responses, in particular the macrophage reaction, and regeneration in Anura is complex. The above-mentioned studies indicate both the positive and negative roles of macrophage reactivity, depending on the development stage and the regeneration model. This dependence has yet to be clarified by a more detailed investigation and comparison of different regeneration models for amphibians.

The interaction of immune system factors with inflammation is critical for the regeneration of various tissues across vertebrates. In a study of factors produced by the immune system and associated with regeneration, special attention was paid to the complement system, plasma proteins that, along with immune cells, play an essential role in the initiation and development of regeneration processes [168,169]. This is complemented by a common conserved effector of the innate immune response, included in the defense system against pathogens [194]. The role of the components of the complement system was considered, using a model for lens regeneration in urodeles [15,17]. C3 and C5 expression turned out to be differential and localized in the area of lens regeneration [195]. The C3 expression was identified in the space of the stroma and epithelium of the dorsal edge of the iris; the C5 expression, in the area of lens and cornea regeneration. The function of the complement proteins C3 and C5 in different Urodela regeneration systems is not yet known. The expression of these molecules was also revealed during the NR regeneration in chicks, which is intriguing [196]. In this model, as in Amphibia, the NR regenerates through the transdifferentiation of RPE and with the involvement of CMZ [197]. The C3a complement was shown to be an inducer of NR regeneration through the activation of the STAT3 transcription regulator, which, in turn, activates the damage response factors IL-6, IL-8, and TNF. This eventually leads to the regulation of the Wnt2b signaling pathway genes, as well as to the expression of the *Six3* and *Sox2* genes characteristic of retinal progenitors [196]. An investigation into the involvement of the immune system’s complement component in maintaining the normal state of the retina in mice showed that the complement activation amplifies the immune responses which, in turn, initiate AMD pathogenesis. Yu and co-authors [198] examined mice that lacked the main complement component C3 and the receptors for complement activation fragments C3a (C3aR) and/or C5a (C5aR). This made it possible to conclude that the C3aR- and C5aR-mediated signaling is necessary to maintain normal retinal function and structure. In general, further studies are required to identify the role of the complement in the NR regeneration/degeneration processes and the differences/similarities between lower and higher vertebrates, as well as between caudate and acaudate amphibians, taking into account species-specificity as well as the specificity related with the age of animals.

### 4.4. Role of Blood Factors and Cells in Initiation of NR Regeneration in Amphibia

It was noted [178] that, in the described models for regeneration-inducing damage in salamanders, particularly as a result of the amputation of a limb and tail, the blood loss was only slight. As is known [199], platelets are mainly involved in blocking blood loss. Amphibians presumably have a high rate of platelet recruitment and delivery to the site of injury for rapid coagulation/hemostasis. These features allow the quick restoration of hemostasis and without the formation of an extensive fibrin clot [199].

Labeling these cells in single-cell sequencing allowed detecting them in a significant amount in the peripheral blood of *Ambystoma* [178]. Platelets are also known as producers of a wide range of cytokines, growth factors, metabolites, and composition modulators of ECM [200]. One of the trigger signals to recovery after the lens removal in the newt is the expression of thrombin [201,202]. It was found that thrombin activation can be detected at the dorsal margin of the iris, the regeneration site, but not at the ventral margin which cannot regenerate the lens. When the inhibitors of thrombin activity were injected into the eye, the dorsal iris cells did not enter the S-phase, and the lens regeneration was inhibited, providing crucial evidence of the vital role of thrombin. It was suggested in [201] that there could be a cellular component-like tissue factor, an integral membrane protein which nucleates the formation of clotting factors that together activate prothrombin. Based on these data, a hypothesis was made about a pattern of early events to stimulate the dedifferentiation and the cell cycle re-entry of iris cells which are, thus, responsible for inducing regeneration. Leukocytes, being attracted by the fibrin clot containing thrombin and the transmembrane protein tissue factor (clotting factor III), activate the expression of FGF2, which, in turn, induces the entry of iris pigmented epithelial cells into the cell cycle [201,203]. It was also suggested [204] that the cell dedifferentiation process in Urodela requires the presence and involvement of thrombin in combination with cellular clotting factors as activators of the process. The role of blood factors and cells in NR regeneration in amphibians requires a dedicated study because of its obvious implication. Due to the role of the immune and circulatory systems in regeneration, the choroid lined by RPE is of particular note, as its presence and regulatory interaction with RPE should mandatorily be taken into account when considering NR factors initiating regeneration. The retina of caudate amphibians does not have an inner vascular membrane. For Anura, the inner vascular membrane of the retina is a substrate for the initiation and development of RPE-derived retinal regeneration, which also suggests the necessity to study the factors produced by it for the initiation of NR regeneration in post-metamorphic frogs.

Similarities in the early injury responses between distantly related species are evident at molecular and cellular levels of organization. However, the divergence in responses even between closely related species becomes more pronounced as diverse cell types are engaged, including those involved in innate immunity. In total, the mechanisms of the cell response largely depend on the contributions of signals from heterogeneous populations of previously differentiated cells that are primarily derived from tissues close to the wound site and signals produced by migrating immune cells [205].

Figure 3 schematically shows a set of events and participants that initiate retinal regeneration in amphibians, as discussed in the sections above and below (Figure 3).

### 4.5. Participants of Molecular Regulatory Networks at the Stage of Initiation of Retinal Regeneration in Amphibians

As a result of various types of damage to the retina or retinal removal in amphibians, various secretory factors are produced by the body and surrounding tissues of the eye. These molecular regulators interact with each other and affect the cell-sources of retina regeneration in amphibians by paracrine or autocrine pathways. The pathways and signaling molecules that regulate the cellular responses to injury and regeneration are still not completely understood. Here, we provide the information concerning the stage of initiation of NR regeneration known to date and obtained on amphibian models. A series of studies [83,84,206] was conducted to elucidate the signaling pathways responsible for the entry of RPE cells into the cell cycle and the transit phase of accumulation of dedifferentiated cells. The authors were based on the assumption that specialized RPE cells, to enter the S-phase, need mitogen-activated protein kinase (MAPK)/extracellular signal-regulated kinase (ERK) kinase (MEK)-ERK intracellular signaling activity, accompanied by the cells’ escape from the stabilizing effects of ECM and the cell-to-cell contacts of RPE. In this study [206], the posterior sector of the eye was cultivated after the isolation of NR for 10 days. This made it possible to reproduce the first stages of NR regeneration similar to those in vivo. The use of this “retinectomy in a dish” showed that MEK–ERK signaling is activated within 1 h post-retinectomy [83] and a number of β-catenin-positive nuclei increases in RPE cells when cell–cell contacts are disrupted by incision. Furthermore, the same result was obtained after the treatment of RPE with ethylene glycol tetraacetic acid (EGTA) [31]. Note that similar results can be obtained on mouse RPE after NaIO_3_ injection [207]. Authors found that oxidative stress induces the dissociation of P-cadherin and β-catenin from the cell membrane and subsequent translocation of β-catenin into the nucleus, resulting in the activation of the canonical Wnt/β-catenin pathway. This indicates the universality of the responses induced by the disruption of intercellular contacts, particularly mitogen-activated protein kinase (MAPK)/extracellular signal-regulated kinase (ERK) kinase (MEK)-ERK intracellular signaling activity, coupled with β-catenin nuclear translocation. Subsequently, this contributes to the transcriptional regulation of injury-responsive genes in RPE cells.

The use of a model with the complete surgical removal of the retina in newts in vivo clarified the timing of the increased activity of MEK–ERK signaling, which turned out to occur as early as 30 min post-retinectomy [84]. However, questions still remain as to what initially regulates the activity of MEK–ERK signaling and whether such regulators are those intracellular and extracellular factors whose release is directly provoked by the operation. A model for the cultivation of the newt eye posterior wall (Sclera + Choroid + BM + RPE) also served to understand the role of the choroid coat as a source of factors necessary for the progression of the proliferative activity of RPE cells [208,209]. FGF2 and IGF-1 were identified as the main triggers of this process.

The role of FGF2 signals in retinal regeneration in vertebrates has been repeatedly demonstrated with various models [16,42,210,211]. Thus, the use of plastic implants containing FGF2 inserted into the eye made it possible to induce the conversion of chicken embryo RPE cells into the retina at stages E22–E24 [212]. A model for retinal removal in chicken embryos at stage E4 showed that, during retinal regeneration, FGF2 acts through the FGFR1,2 receptors, stimulating the phosphorylation of Erk. This chain of events depends on the activity of the MEK substrate, thereby demonstrating that RPE transdifferentiation in birds occurs through the Fgf-Fgfr-MEK–ERK signaling cascade, whose activation is also associated with the up-regulation of the TF Pax6 [213]. We also conducted a study of the localization and expression of components of the FGF2 signaling pathway in the posterior wall tissues of a normal eye and an eye of an adult newt, *P. waltl*, wherein retinal regeneration was induced. The expression of the *fgf2*, protein Fgf2 and its receptors Fgfr2, which have a high degree of homology with those found in other vertebrates, was detected in the NR, ON, CMZ, RPE, and choroid of the native eye. RT-PCR, as well as immunohistochemistry data, showed the down-regulation of the *fgf2* gene and FGF2 ligand expression compared to those in the same tissues of the non-operated eye in the early period after retinal removal (4–8 days) [214]. In a study of the molecular profile of gene expression using a microarray, a decrease in the expression levels of genes from the FGF family (*fgf1*, *fgf2*, and *fgf4*) and genes encoding receptors (*fgfr1*, *fgfr2*, and *fgfr4*) was observed in a number of different regenerating systems of newt [215]. The authors explained this phenomenon by the fact that, after surgery, the proteins to be used first are Fgf, previously synthesized and accumulated in the ECM and released as a result of its remodeling at the initial regeneration stage. We also assume that the above-discussed factors involved in the mechanisms of RPE cell protection from death are in demand to a greater extent in the tissues of the posterior wall of the eye during the early period. For amphibians and other vertebrates, another such protection factor is the epidermal growth factor (EGF) [216]. As regards Fgf2/*fgf2*, it is obvious that the mitogenic effect of this factor later becomes manifest during the period of the increase in the proliferative activity of cell sources of regeneration [217]. This opinion was also expressed by other researchers who suggested that FGF2 is not the primary one inducing the RPE conversion process during NR regeneration [206].

Thus, only some links of the regulatory network, which is a trigger of regeneration, are known, but their relationships and the order of events should be investigated in the future. As a result of a transcriptomic study, these trigger events were found to be associated with a decrease in the expression of genes characterizing the initial differentiation of RPE cells such as *RPE65*, *CRALBP/RLBP1*, *ZO1*, *Mitf*, *Otx2*, and *Musashi1a/c*, whereas genes associated with cell-cycle progression such as *Cyclin D1*, *CDK4*, *Histone H3*, and those associated with growth factor signaling such as *FGFR1* and *FGFR3* are upregulated [148].

The activity of immune response genes and proto-oncogenes *c-fos*, *c-myc*, *c-jun* is further manifested in the early phase of RPE reprogramming [218]. As was observed in experiments with isolated cells using q-PCR, the first daughter RPE cells at the beginning of retinectomy-induced proliferation demonstrated the expression of some pluripotency genes, *c-Myc*, *Klf4*, and *Sox2*, and, along with them, “developmental” *Mitf* and *Pax6* [32]. RPE cells, which newly expressed *c-Myc*, *Klf4*, *Sox2*, *Mitf*, and *Pax6*, retained a sign of melanogenic differentiation, the expression of RPE65, as was observed for both the mRNA and protein levels. It was assumed that the pluripotency factors c-Myc, Klf4, and Sox2 are capable of reprogramming and even initializing RPE cells, as demonstrated in in vitro studies on mammals [219]. The aforementioned differential expression of the *Pax6* gene is considered coupled with the expression of early trigger regulatory TFs and signaling systems [38]. The authors managed to identify four classes (v1, v2, v3, and v4) of Pax6 transcription variants in the eye of adult newt, *Cynops pyrrhogaster*. In the study, their expression could be detected in many tissues of the non-operated eye, but not in tissues of the posterior wall of the eye. After performing retinectomy in in vivo and in vitro systems, Inami with co-authors [38] observed the initiation of the expression of the first and second variants of the Pax6 gene. Of certain importance is also the observation of the independent upregulation of *Pax6* variants on the MEK–ERK pathway, but related to others, not identified signaling pathways [38].

The upregulation of the expression of the pro-neural factor N-Notch also does not look like a regeneration-initiating event. When N-Notch hybridization was used in situ, the signal was observed no earlier than in early proliferating progenitors of the regenerating retina of an adult newt, in a transit double-row neuroblast cell population [220,221]. However, Nakamura and Chiba, based on PCR data on the presence of Hes-1, neurogenin1, and also Delta-1 expression in adult newt RPE cells, assumed that the Notch signaling pathway was switched on at earlier stages of the RPE conversion process [221]. In general, the molecular basis of the retinal regeneration initiation in Urodela requires further study by applying new methods and approaches that have become available thanks to the increasingly accumulated information on the genomes of some model species of this amphibian order [14,222,223,224].

Substantial effort have been made to identify the participants of the regulatory network responsible for initiating retinal regeneration in Anuran amphibians. The studies by [21], using *Xenopus* with surgically excised retinas, showed the role of ERK1/2-mediated retina regeneration induced by exogenous FGF2. As mentioned above, after retinectomy in these animals, RPE cells migrate to the RVM, which facilitates the RPE transdifferentiation, although the inducing factors and molecular mechanisms present in the RVM are not fully understood. However, retinectomy that also includes the removal of the RVM, followed by FGF2 induction, successfully results in a regenerated retina, indicating that FGF2 is sufficient in this regard [21]. FGF2 activates the MEK–ERK pathway and, similarly to the newt NR regeneration [83], the MEK–ERK pathway is activated during the early stages of the regenerative process, while an inhibition of MEK signaling significantly impairs NR regeneration in frogs [21]. These observations once again emphasize an important if not key the role of MEK–ERK activity in the modulation of RPE cells’ regenerative responses [225]. However, it was shown [226] that Pax6 alone is sufficient to induce the RPE cell transdifferentiation without the addition of FGFs or surgical manipulation. It turned out that the Pax6-mediated RPE conversion can be induced even at later stages of development in *Xenopus*. Both in vivo and in vitro studies showed that the Pax6 lies downstream of FGF signaling. The authors [226] suggested that TF Pax6 plays central roles in the regulatory network responsible for RPE transdifferentiation. Nabeshima et al. [160] set up experiments using a transgenic *Xenopus* line, in which the EGFP expression was under the control of the *Rax* promoter. The *Rax-EGFP* expression was analyzed during NR regeneration in a tissue culture model based on F1 and F2 generations. the expression of TFs such as Pax6 and Rax-EGFP was observed, with the *Rax-EGFP* expression preceding the *Pax6* expression, and with the expression of both genes occurring in RPE cells that had lost contact with the BM facing the choroid. Subsequently, the upregulation of the *Pax6* and *Rax* genes was recorded during the process of RPE conversion [69,160].

The regulatory mechanisms governing the behavior of CMZ cells during NR regeneration in tailless amphibians at postembryonic stages of development were also discussed. One of the key components of this regulatory network, according to [227], is the canonical Wnt signaling pathway. Using a GFP-based Wnt-responsive reporter, the authors showed that the canonical Wnt signaling in CMZ is activated in transgenic *Xenopus* tadpoles. To clarify the regulatory role of Wnt signaling, transgenic, hormone-inducible canonical Wnt pathway-activating and repressing systems were produced, and directed to specifically intersect at the nuclear endpoint of transcriptional Wnt target gene activation. The postembryonic induction of the canonical Wnt pathway in transgenic animals was found to result in an increase in the pool of proliferating cells in the CMZ. The authors attributed these observations to a delay in the exit of CMZ cells from the cell cycle. These data were confirmed by the results a pulse-chase experiment on BrdU-labeled retinal precursors. Conversely, repression of the canonical Wnt pathway inhibited the proliferation of CMZ cells [227].

Hedgehog signaling, whose involvement is shown in the development of the *Xenopus* eye, is a probable regulator of the CMZ cells’ behavior during NR regeneration in acaudate amphibians [228]. The authors showed that the role of Shh consists of accelerating the cell cycle in the developing retina by reducing the length of the G_1_ and G_2_ phases. Conversely, retinal progenitors with the blocked Hedgehog signaling cycle are slower, with longer G_1_ and G_2_ phases, and remain in the cell cycle longer. It was assumed that Hedgehog may modulate cell cycle kinetics through the activation of the key cell cycle activators cyclin D1, cyclin A2, cyclin B1, and cdc25C. An assumption can be made that, as in the development of the retina of the eye, competitive, antagonistic interactions of Wnt and Shh signaling pathways are possible during retinal regeneration in Anura [229]. Thus, the data on the regulatory networks involved in the initiation and progress of NR regeneration in Anura correlate with and complement the data obtained for Urodela retinal regeneration models. It is obvious that the differences in cell response are preset in the genetic program of the source cells of regeneration proper and are conditioned by the epigenetic regulation that underlies the commitment of retinal progenitor cells in settings of retinal development and regeneration. Epigenetic modifications and regulatory mechanisms at both DNA and chromatin levels are also postulated to play an important role in the timing of the differentiation of specific retinal cells [230].

The above-described early events in amphibians, associated with the separation of RPE and NR, can be regarded as common responses to cell damage and cell stress. Despite the significance of understanding injury as an important determinative factor, the evolutionary role of characteristics of injury and cell response mechanisms remains a topic that needs more focused research. With the wide range of retinal responses caused by tissue damage such as cellular and oxidative stress, the suppression of initial biochemical reactions, the work of cell protection factors, and the rearrangement of the environment, amphibians have evolutionarily fixed features [93]. These features ultimately determine and, to a high degree, represent the regulatory mechanisms, not only contributing to but also being involved in the program for the implementation of NR regeneration in amphibians.

## 5. Conclusions

Amphibians are the only class of tetrapods exhibiting the highest potential to regenerate the retina of the eye after damage, even in the case of its removal at mature age. For this reason and taking into account the high demand for the treatment of degenerative diseases of the human retina, the examples of retinal regeneration in amphibians have been studied for a long time and from many aspects. The mechanisms initiating the process of retinal regeneration in amphibians are triggers of activation of regeneration source cells (RPE and CMZ), their proliferation, and conversion into neural cells and differentiation into the cell types of the NR forming de novo. The initiation of these processes depends on the characteristics of the injury inflicted and a vast range of subsequent events (Figure 3). Their set includes local changes occurring in the system: choroid, BM, RPE, NR, as well as systemic responses from the immune and circulatory systems. The objective of this review was an attempt to identify the features in this set that are characteristic of retinal regeneration-competent amphibians. One such feature is the relatively high viability of regeneration source cells, which is presumably the result of a low rate of cellular metabolism and support from viability factors that not only compensate for cellular stress but also contribute to the progress of regeneration. Oxidative stress can also be part of the protective mechanism and, simultaneously, a participant in the general regulatory system that initiates NR regeneration in amphibians. When the antioxidant protection is considered, both conservatism and variability are assumed. The latter is manifested as the range of molecular participants of signaling pathways associated with maintaining homeostasis and providing endogenous cell protection.

Another feature is ECM remodeling, in which changes in the expression of fibronectin, laminin, tenascin, N-CAM, matrix metalloproteases, etc. were observed. In parallel, changes in the cell surface and the structure of intercellular contacts occur, particularly in the β-catenin–cadherin complex. For amphibians, these events are considered as inextricably linked with the initiation of RPE proliferation via the MEK–ERK signaling pathway, and then promoted through FGF. A rapid rearrangement of the RPE cytoskeleton and the transition from epithelial-to-neural specificity were immediately recorded after the separation from NR. In Urodela, such a rearrangement, along with information about the change in the gene expression pattern characteristic of melanogenic and epithelial differentiation into that specific for neural progenitors, indicates a state of the gene expression regulation system permissive of RPE cell conversion. The success of retinal regeneration in amphibians is assumed to be genetically inherent and determined by the specifics of epigenetic regulation, which provides them with regenerative advantages. The study of the epigenetic landscape and the changes in chromatin organization during the initiation of retinal regeneration in amphibians constitute one of the most urgent research topics [231].

The success of retinal regeneration as well as other tissues in amphibians is determined by a set of interactions between cells that are sources of regeneration, on the one hand, and cells and factors of the “primitive” immune system, on the other. In these interactions, macrophages that are characterized by morpho-functional heterogeneity play a special, multilateral role; however, differing between Anura and Urodela. The significance of this role is noted at different stages of NR regeneration; during the initiation of retinal regeneration in Urodela, it is detected while RPE cells are freed of the initial features, with changes in the cell surface and in the formation of the pro-regenerative environment. A feature of the initiation of NR regeneration in amphibians is also the rapid coagulation hemostasis provided by platelet recruitment and by the involvement of blood factors (complement, thrombin, and tissue factor) that were found in Urodela during lens regeneration, which is a different regeneration system but close to the retinal one. The role of cells of the immune and circulatory systems and the regulatory factors produced by them in the NR regeneration in amphibians are among the most important fields for further research. In general, the main feature of NR regeneration triggers in amphibians is that they are not only a multidimensional response to damage but they also create a permissive environment allowing the source cells to activate and rebuild the retina de novo. This circumstance is a key to finding approaches to triggering the retinal regeneration in mammals and humans.

The data on intracellular and extracellular regulatory factors initiating the NR regeneration, collected to date, along with the extremely essential data from further molecular studies, contribute to addressing a number of important issues. This facilitates a deeper understanding of the outline of the regulatory network for the progress of NR regeneration including, in particular, the processes of the reprogramming and proliferation of source cells, RPE and CMZ. The definitive goal of this field of research is an attempt to translate data into practical ophthalmic medicine for addressing the issues related to the poorly treatable degenerative diseases of the human retina.

## Figures and Tables

**Figure 1 life-13-01981-f001:**
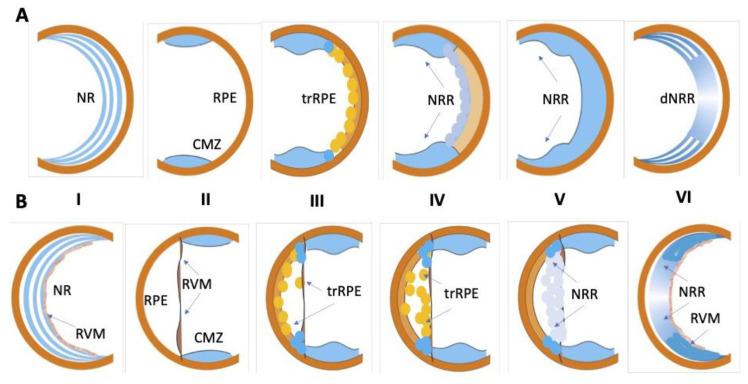
Schematic diagram of retinal regeneration in amphibians. (**A**)—Urodela (newt): I—intact differentiated neural retina (NR) by attached retinal pigmented epithelium (RPE); II—NR is removed, ciliary marginal zone (CMZ) and RPE after retinectomy. III—RPE cells transdifferentiate (trRPE), CMZ cells grow; IV,V—NR regenerate (NRR) forms and VI—differentiates (dNRR). (**B**)—Anura (Xenopus): I—intact differentiated neural retina (NR) with retinal vascular membrane (RVM) and retinal pigmented epithelium (RPE); II—NR is removed, CMZ, RPE and RVM are preserved; III—RPE cells leave RPE layer, migrate towards RVM and transdifferentiate (trRPE); IV—RPE cells setting on the RVM proliferate and transdifferentiate (trRPE); V,VI—RPE cells on the RVM proliferate and continue to reprogram into neural retinal cells, whose population subsequently form neural retina regenerates (NRRs).

**Figure 2 life-13-01981-f002:**
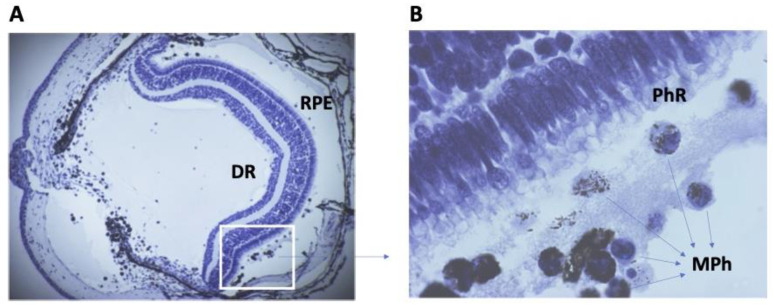
Cross-section of the eye of the newt after lens removal and retinal detachment. (**A**)—General view. RPE—retinal pigmented epithelium; DR—detached retina; Magnification: 40×. The area in the white box in (**B**). (**B**)—the space between photoreceptors (PhR) of DR and RPE is filled with phagocytic cells, monocytes, and macrophages (MPh). Most macrophages are filled with pigment granules. Magnification: 1000×.

**Figure 3 life-13-01981-f003:**
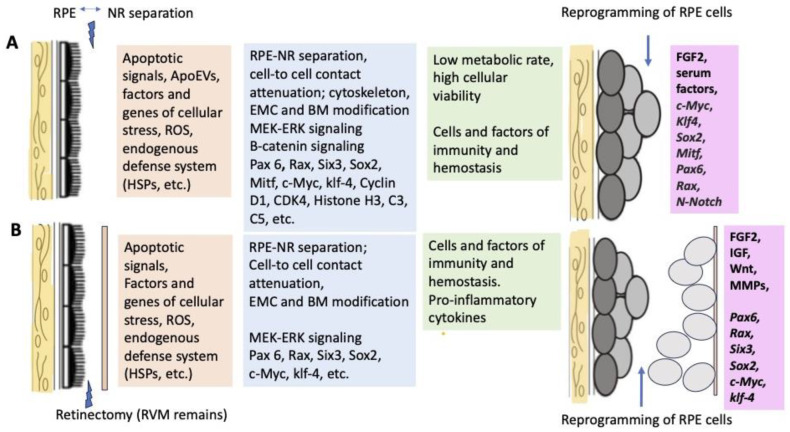
A range of known cellular and molecular factors involved in the initiation of retinal regeneration and RPE reprogramming in Urodela (**A**) and Anura (**B**).

**Table 1 life-13-01981-t001:** Ways to damage the retina to induce its regeneration in Urodela and Anura.

Amphibians/Age	Type of Damage	Regeneration,Cell Sources	Cell Source Responses, NR Regeneration Steps
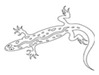 NewtMature	Retinal removal	YesRPE,CMZ	CMZ cell proliferation and differentiation; RPE cell transdifferentiation,proliferation, retinal anlage formation, retinogenesis [25,28,58]
Optic nerve and blood vesselscrosscut	YesRPE,CMZ	Degradation of the initial retina, cleaning of its remnants; CMZ cell proliferation and differentiation; RPE cell transdifferentiation, proliferation, retinal anlage formation, retinogenesis [59,60]
Retinal surgical detachment	YesRPE,CMZ,Bipolar-like cells	Cell death in the outer retina, RPE cell transdifferentiation, proliferation, CMZ cell proliferation and differentiation. Possibility of retinal reattachment [53,61]
Long-term illumination	Yes,Bipolar-like cells, RPE cells	Replacing some dead photoreceptors by displaced bipolars and migrating RPE cells [63]
NewtMature	Transplantation of the eye posterior wall without retina into lensectomized eye cavity or culturing in vitro	Yes(NR fragment),RPE	RPE cell transdifferentiation, proliferation, retinal anlage (fragment) stratification [64,82,83,84]
* 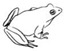 *Frog/mature, tadpoles*Xenopus laevis*	Retinal removal, RVM remaining	Yes,RPE, CMZ	RPE cell migration to the RVM, transdifferentiation, proliferation, RPE-derived retinal anlage differentiation; CMZ cell proliferation [20]
*Xenopus laevis*	Isolation of posterior wall tissues of the eye and culturing in vitro	Yes,(NR fragment), RPE	RPE transdifferentiation,proliferation, retinal anlage (fragment) differentiation [20]
*Xenopus tropicalis*	Retinal removal	Yes,CMZ	CMZ cell proliferation and differentiation [43]
Transgenic*Xenopus laevis*	Chemogenetic models of photoreceptor ablation	Partial NR regeneration, CMZ,cells of the inner retina	Photoreceptor cell replacement[74,75,76]
Frog/tadpoles *Rana catesbiana*	Retinal devascularization	Yes,CMZ, RPE	RPE cell conversion, CMZ cell proliferation and differentiation [67,68]

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
