# Peer review of "Cellular and Molecular Triggers of Retinal Regeneration in Amphibians"

_life, 2023, doi:10.3390/life13101981_

Round 1

Reviewer 1 Report

Abstract:

Line 18: It is the first time you show ECM so it should be written like   Extracellular matrix (ECM).

It would be nice if in the text you explain "the difference" between CMZ and GMZ. The first time that you name GMZ is in a figure caption.

Recommendation: reference papers always the same way. Choose between numbers or name of authors + number of reference.

Recommendation: It would be nice if you add a figure where you can differentiate into the three the different processes and sources of new cells (RPE, CMZ and Müller cells)

Line 214: "Martínez-DeLuna" instead of "Martinez-DeLuna"

Line 244: in situ should be italic

Table 1:    First line of the table -- cell source: "formation of retinal anlage"

                  Suggestion-- related to (?) simbol in the table: I would add in the table caption what that means (there are other processes not well known)

                  "Transplantation of the eye posterior wall..." belong to newt or frog?

                   From Xenopus tropicalis in advance, the table is a bit confusing: Chemogenetic models belongs to...?

Sometimes the authors write 3-D and some other 3D. It would be nice if you can always write it the same way.

Line 405: BM should be abbreviated for first time.

Line 58, 404 and 450: the authors have already abbreviated ECM in line 58. It is not needed to point what is the meaning of the abbreviation again in line 450 (the authors use the abbreviation in line 404).

Line 286 and 554: in line 286 ROS is the first time that should be abbreviated, not in 554.

Line 811: in vivo should be italic.

Line 993: reference

Conclusions:

I miss that the authors hypothesize about how all these information can be translated into practical ophtalmic medicine.

Author Response

Reviewer 1.

Dear reviewer 1. Foremost, we are very grateful for the review, comments made and pointing out the omissions.

Comments and Suggestions for Authors

1). Line 18: It is the first time you show ECM so it should be written like Extracellular matrix (ECM). The abbreviation is revealed

2). It would be nice if in the text you explain "the difference" between CMZ and GMZ. The first time that you name GMZ is in a figure caption.

Growth marginal zone (GMZ) is replaced by ciliary marginal zone (CMZ) for uniformity of names of the same area (through out of the text)

3). Recommendation: reference papers always the same way. Choose between numbers or name of authors + number of references.

Most of authors’ names were removed throughout the text, but kept in lines 432,214,726 and 906 to save the sentences readable 

4). Recommendation: It would be nice if you add a figure where you can differentiate into the three the different processes and sources of new cells (RPE, CMZ and Müller cells)

In the article we write that the question of Muller cells’ participation in retinal regeneration in amphibians has not been sufficiently studied and is far from being resolved. Moreover, as we write in the review, the participation of distinct cell sources, their share in retinal regeneration may depend on the animal age, amphibian species, type of injury, etc. It seems difficult to schematically depict all these differences. The manuscript consists Fig.1 and the table which present a scheme of the process and participation of cell sources in it in Urodela and Anura.

5). Line 214: "Martínez-DeLuna" instead of "Martinez-DeLuna". The spelling of the name has been corrected.

 6). Line 244: “in situ” should be italic. In accordance with the accepted for papers in journals of MDPI, italic is not used for words “in vivo, in vitro, in situ”.

 7). Table 1:    First line of the table -- cell source: "formation of retinal anlage". To be more correct “NR regeneration steps”- added. 

 8). Suggestion-- related to (?) simbol in the table: I would add in the table caption what that means (there are other processes not well known).

Unfortunately, I didn't quite understand this recommendation. A detailed description of the processes is given in the text.

9). "Transplantation of the eye posterior wall..." belong to newt or frog? It was updated – mature newts – added to the table

 10). From Xenopus tropicalis in advance, the table is a bit confusing: Chemogenetic models belongs to...?

Тhe lines in the table have been corrected. The problem is that they shift when transferring from one computer to another.

11). Sometimes the authors write 3-D and some other 3D. It would be nice if you can always write it the same way. Should be “3D” – corrected.

12). Line 405: BM should be abbreviated for first time. Abbreviation BM – revealed.

13). Line 58, 404 and 450: the authors have already abbreviated ECM in line 58. It is not needed to point what is the meaning of the abbreviation again in line 450 (the authors use the abbreviation in line 404). It was corrected – “extracellular matrix” (line 450) deleted.

14). Line 286 and 554: in line 286 ROS is the first time that should be abbreviated, not in 554. –It was corrected.

15) Line 811: in vivo should be italic. In accordance with the accepted for words “in vivo, in vitro, in situ” italic is not used for papers in MDPI journals,.

16) Line 993: reference. It was corrected: (Jia et al., 2023) - deleted

17). Conclusions:

I miss that the authors hypothesize about how all these information can be translated into practical ophthalmic medicine.

Translation to practical medicine of findings described in our review has certain perspectives since the main key triggers for retinal regeneration are becoming increasingly clear. To stress it out we added additional sentences to the “Conclusion”: “In sum, the main feature of NR regeneration triggers in amphibians is that they are not only a multidimensional response to damage. They also create a permissive environment allowing the source cells to activate and rebuild the retina de novo. This provision is key to finding approaches for triggering retinal regeneration in mammals and humans”.

Reviewer 2 Report

This paper is a review article covering retinal regeneration in amphibians, which is an interesting topic.

-The abstract is written like an incomplete introduction section, rather than summarizing the significance and content included in the manuscript.

-Given that many of the mechanisms of regeneration are conserved in vertebrates, it is still unclear from this review why certain amphibians are able to regenerate and others not. Figure 3 provides a summary of the pathways affected, but many of these pathways are involved in tissue wound healing in mammals as well.

-The IRB statement (lines 1021 - 1023) does not appear relevant to this review article.

Minor editing is needed.

Author Response

Reviewer 2

 Authors of the manuscript express their gratefulness to the Reviewer 2 for careful reading and comments made. We have taken them all into account (below).

Comments and Suggestions for Author:

This paper is a review article covering retinal regeneration in amphibians, which is an interesting topic.

  1. -The abstract is written like an incomplete introduction section, rather than summarizing the significance and content included in the manuscript.

We have tried to devote most of the abstract (lines 15-23) to the issues covered in the review and concerned the main theme: mechanisms of initiation of retinal regeneration in amphibians. Lines 8-14 give a necessary preamble to this. Detailing the features of all NR regeneration triggers in amphibians is impossible within the 200 words of the abstract.

  1. -Given that many of the mechanisms of regeneration are conserved in vertebrates, it is still unclear from this review why certain amphibians are able to regenerate and others not.

We simply have nothing to build on to discuss the ability/inability of various amphibian species to regenerate the retina of the eye. All the species of Urodela amphibians studied at different times by different research teams (caudate amphibians: Triturus viridescens, Triturus cristatus, Pleurodeles waltl, Notophtalmus viridescens, Ambystoma mexicanum, and Anura: Xenopus laevis, Rana catasbiana, and Rana tropicalis) demonstrated the ability to regenerate the retina of the eye. The major features of regeneration observed within each of these two orders were the NR regeneration rate and the different contributions made by the cell sources, RPE and CMZ. – these sentences have been added to the text (lines 46-51).

  1. -Figure 3 provides a summary of the pathways affected, but many of these pathways are involved in tissue wound healing in mammals as well.

Yes, this is so. These findings emphasize once again the universality of the spectrum of cellular and molecular events which occur as a result of RPE and NR separation after injury or in disease. This spectrum is indicated in the manuscript. However, even inside these events there are features we tried to highlight. In sum, in amphibians, everything that happens after injury creates an environment permissive for gene expression, leading to conversion and proliferation of source cells, and retinal histogenesis de novo. This is mentioned in blue and rose columns of the picture 3.

-The IRB statement (lines 1021 - 1023) does not appear relevant to this review article.

I didn't quite understand if the question is addressed to authors or the editor. Is it a question of the need to provide this information for a review article, or in essence of what is written …? Indeed, Markitantova Yu. and Grigoryan E., working long time in the same laboratory and being close colleagues, prepared an article jointly.

Comments on the Quality of English Language

Minor editing is needed – small mistakes have been corrected.

Reviewer 3 Report

A better mechanistic understanding of retinal regeneration in amphibians can provide insights into potential treatments for retinal degeneration in humans. The review article “Cellular and Molecular Triggers of Retinal Regeneration in Amphibians” written by Markitantova et. al. is well written and exhaustive. However, I have few minor concerns.

1.    At few places, the sentences are too lengthy making it difficult to follow the intended message. For example – line 427-430 – “Hasegawa and co-authors...”, and  line 633-636 – “A study of the process ...”

2.    Some of the statements are confusing. For example – line 611-612 – “Therefore, of particular surprise...” and line 627-629 – “It is assumed that....”

3.    Line 964 – “and conversion into neural cells (RPE)” should not be RPE.

Author Response

Reviewer 3

First of all, we would like to express our gratitude to the reviewer for his(her) attentive attitude to the text and recommendations for its correction (below).

Comments and Suggestions for Authors

A better mechanistic understanding of retinal regeneration in amphibians can provide insights into potential treatments for retinal degeneration in humans. The review article “Cellular and Molecular Triggers of Retinal Regeneration in Amphibians” written by Markitantova et. al. is well written and exhaustive. However, I have few minor concerns.

  1. At few places, the sentences are too lengthy making it difficult to follow the intended message. For example – line 427-430 – “Hasegawa and co-authors...”, and line 633-636 – “A study of the process ...”

Lines: 427-430, 633-636: These long sentences were divided into two shorter. It is marked on the text right margin.

  1. Some of the statements are confusing. For example – line 611-612 – “Therefore, of particular surprise...” and line 627-629 – “It is assumed that....”

Line 611-612.  The sentence has been rewritten: “Taking into account the existence of blood-retina barrier, it seems interesting and unusual the involvement of newly coming inflammatory cells in addition to resident ones in retinal regeneration in Urodela”.

Line 627-629. – The sentence has been rewritten: “It is assumed that non-dividing macrophages arrive in the site of damage where they and resident macrophages show the ability to actively reproduce”.

  1. Line 964 – “and conversion into neural cells (RPE)” should not be RPE. (RPE) - deleted.

Round 2

Reviewer 2 Report

Most of the issues were addressed in the response. The text is still difficult to read in places and the abstract should include a sentence to summarize the purpose of the review.

Minor edits were made.

Author Response

Dear Reviewer 2, thank you for your notices. Below is a new version of the abstract. Last two sentences were added in order to show the main purpose of the review (italic).

Understanding of the mechanisms triggering initiation of retina regeneration in amphibians may advance the quest for prevention and treatment options for degenerating human retina diseases. Natural retina regeneration in amphibians requires two cell sources, retinal pigment epithelium (RPE) and ciliary marginal zone. Disruption of RPE interaction with photoreceptors through surgery or injury triggers local and systemic responses for retinal protection. In mammals, disease-induced damage to the retina results in shutdown of function, cellular or oxidative stress, pronounced immune response, cell death and retinal degeneration. Differently from retinal pathology in mammals, the regenerative responses in amphibians have taxon-specific features ensuring efficient regeneration. They include rapid hemostasis, recruitment of cells and factors of endogenous defense systems, activities of the immature immune system, high cell viability, and efficiency of extracellular matrix, cytoskeleton, and cell surface remodeling. These reactions are controlled by specific signaling pathways, transcription factors and epigenome, which are insufficiently studied. This review provides a summary on the mechanisms initiating retinal regeneration in amphibians and reveals its features collectively directed at recruiting universal responses to trauma to activate cell sources of retinal regeneration. The study of the integrated molecular network of these processes is a prospect for future research in demand biomedicine.

We have also divided several long sentences into two to make reading easier. These sentences were marked on the right margin of the manuscript.
